# Necrotic Cell Death and Inflammasome NLRP3 Activity in *Mycobacterium bovis*-Infected Bovine Macrophages

**DOI:** 10.3390/cells12162079

**Published:** 2023-08-17

**Authors:** Omar Escobar-Chavarría, Alejandro Benitez-Guzman, Itzel Jiménez-Vázquez, Jacobo Carrisoza-Urbina, Lourdes Arriaga-Pizano, Sara Huerta-Yépez, Guillermina Baay-Guzmán, José A. Gutiérrez-Pabello

**Affiliations:** 1Departamento de Microbiología e Inmunología, Facultad de Medicina Veterinaria y Zootecnia, Universidad Nacional Autónoma de México, Mexico City 04510, Mexico; omare@fmvz.unam.mx (O.E.-C.); alebenitezg@fmvz.unam.mx (A.B.-G.); itzalx1402@hotmail.com (I.J.-V.); jacobourbina@yahoo.com.mx (J.C.-U.); 2Unidad de Investigación Médica en Inmunoquímica, Hospital de Especialidades del Centro Médico Siglo XXI, Instituto Mexicano del Seguro Social, Mexico City 06720, Mexico; landapi@hotmail.com; 3Unidad de Investigación en Enfermedades Oncológicas, Hospital Infantil de México, Federico Gómez, Mexico City 06720, Mexico; shuerta@yahoo.com (S.H.-Y.); guillebaay@gmail.com (G.B.-G.)

**Keywords:** bovine tuberculosis, inflammasome, cell death pathways, pyroptosis, ASC specks, caspase-1

## Abstract

*Mycobacterium bovis* is a facultative intracellular bacterium that produces cellular necrosis in granulomatous lesions in bovines. Although *M. bovis*-induced inflammation actively participates in granuloma development, its role in necrotic cell death and in bovine macrophages has not been fully explored. In this study, we evaluate the effect of *M. bovis* AN5 and its culture filtrate protein extract (CFPE) on inflammasome activation in bovine macrophages and its consequences on cell death. Our results show that both stimuli induce necrotic cell death starting 4 h after incubation. CFPE treatment and *M. bovis* infection also induce the maturation of IL-1β (>3000 pg/mL), oligomerization of ASC (apoptosis-associated speck-like protein containing CARD), and activation of caspase-1, following the canonical activation pathway of the NLRP3 inflammasome. Inhibiting the oligomerization of NLRP3 and caspase-1 decreases necrosis among the infected or CFPE-stimulated macrophages. Furthermore, histological lymph node sections of bovines naturally infected with *M. bovis* contained cleaved gasdermin D, mainly in macrophages and giant cells within the granulomas. Finally, the induction of cell death (apoptosis and pyroptosis) decreased the intracellular bacteria count in the infected bovine macrophages, suggesting that cell death helps to control the intracellular growth of the mycobacteria. Our results indicate that *M. bovis* induces pyroptosis-like cell death that is partially related to the NLRP3 inflammasome activation and that the cell death process could control bacterial growth.

## 1. Introduction

*Mycobacterium bovis* (*M. bovis*), the causal agent of bovine tuberculosis (bTB), is an intracellular bacterium that causes heavy economic losses in bovine livestock production and represents a public health problem. *M. bovis* infects numerous mammals, including cattle, goats, dogs, pigs, lions, elephants, badgers, possums, deer, bison, and humans. The effects of *M. bovis* infection are variable; while recent infections do not induce notable changes, animals with chronic infection may develop weight loss, loss of appetite, and weakness, especially zoo animals [1]. In cattle, infection occurs mainly through the inhalation or ingestion of the bacterium, and severe infection causes loss of appetite and weight, and decreases productive parameters [2,3].

One of the main zoonotic transmission pathways of bovine tuberculosis to humans is by consuming unpasteurized dairy products, which generates extrapulmonary tuberculosis [4]. However, people infected with *M. bovis* by the respiratory route show lesions similar to those caused by *Mycobacterium tuberculosis* [5]. In this host–pathogen interaction, bacteria fight to survive, and the immune system tries to resolve the infection by accumulating inflammatory cells in the lung, mainly macrophages, epithelial-like macrophages, multinucleated giant cells, and lymphocytes. This accumulation generates a chronic lesion known as the granuloma. In cattle, lung and lymph node granulomas are similar but have small differences; for example, lung granulomas, but not lymph node granulomas, display neutrophil infiltration. The granuloma is a collection of extensive necrosis, mineralization, and abundant cell infiltration delimited by a connective tissue capsule. However, it is still unclear how species of the *Mycobacterium tuberculosis* complex induce necrosis. Therefore, studies that characterize these mechanisms are required.

Necrosis is a cell death process characterized by the destruction of functional tissue. Cells lose membrane integrity, allowing the escape of cellular components that initiate an inflammatory process [6,7]. For many years, necrosis was considered accidental, but it is now clear that necrotic cell death is a complex phenomenon involving numerous mechanisms. Further studies have classified it into necroptosis, pyroptosis, and ferroptosis [6,7,8,9].

Pyroptosis is a type of regulated necrotic death mediated by the inflammasome, which consists of multiprotein structures with several components: a receptor of either the NLR (nucleotide-binding oligomerization domain-like receptors) or the AIM myeloma 2 (AIM2)-like receptor family, the ASC (apoptosis-associated speck-like protein containing a CARD), and pro-caspase-1 [6,10]. The main role of inflammasomes is the maturation of cytokines of the IL-1 superfamily (IL-1β, IL-18, and IL-33), but they can also cleave the pore-forming molecule gasdermin D [7,8]. The pores formed by gasdermin D promote IL-1β release and generate an osmotic imbalance that leads to cell death by lysis [7,8].

Many bacteria of the genus *Mycobacterium* activate the inflammasomes NLRP3, NLRP7, and AIM2 in mouse and human macrophages, which in some cases induce pyroptosis and subsequent tissue damage [9,11]. However, the role of *M. bovis* in inflammasome activation that may lead to necrotic cell death in bovine macrophages has not been fully explored. This study demonstrates that incubating bovine macrophages with *M. bovis* AN5 or its culture filtrate protein extract (CFPE) induces IL-1β production mediated by the NLRP3-ASC-caspase-1 axis and pyroptosis-like necrotic cell death.

## 2. Materials and Methods

### 2.1. Bacteria

This study used *M. bovis* AN5, an international reference strain that exhibits intermediate virulence, in the mouse model used [9]. This strain was originally isolated in England around 1948 and used to produce the bovine-purified protein derivative (PPD) applied in the tuberculin test. All bacteriological procedures were performed in a biosafety unit. Bacteria were grown at 37 °C in shaking conditions and shaken in Middlebrook 7H9 broth (Becton Dickinson, Cockeysville, MD, USA) with 0.05% Tween 80 and 10% oleic acid—albumin—dextrose—catalase (OADC) enrichment. The bacterial suspension was obtained by centrifuging cultures at 2500× *g* for 10 min and resuspending the pellet in RPMI (Gibco, Life Technologies, Grand Island, NE, USA). One-milliliter aliquots were stored at −80 °C, and the inoculum was titrated by plating serial dilutions on Middlebrook 7H11 medium with 10% oleic acid-albumin-dextrose-catalase (Becton Dickinson, Pottery Road, IE, USA).

### 2.2. Culture Filtrate Protein Extract (CFPE)

The CFPE was obtained by incubating *M. bovis* strain AN5 in Sauton Medium enriched with 0.5% glycerol (as the sole carbon source) and 4 g/L of L-asparagine for eight weeks at 37 °C without shaking. After the incubation, the bacterial biomass was filtered using 0.45 μm and 0.22 μm pore membranes (Merk Millipore, Tullagreen, IE, USA). The CFPE was then precipitated with 0.5 g/mL of 70% ammonium sulfate.

### 2.3. Bovine Macrophage Culture

Macrophages were obtained from peripheral blood mononuclear cells (PBMCs) according to a method previously described by our laboratory [12,13]. Briefly, peripheral venous blood was collected in acid citrate dextrose (ACD) from healthy adult cattle and centrifuged at 1000× *g* for 30 min at room temperature. The buffy coats were suspended with an equal volume of citrated PBS pH 7.4, layered onto 15 mL of Histopaque (Sigma Aldrich, Saint Louis, MO, USA) with a specific density of 1.077, and centrifuged at 1200× *g* for 20 min. PBMCs were then removed from the interface between the plasma and histopaque solution, pooled, diluted in 50 mL of citrated PBS, and centrifuged at 500× *g* for 10 min. The cell pellets were washed thrice with citrated PBS at 500× *g* for 10 min, suspended in RPMI 1640 with L-glutamine, amino acids, sodium pyruvate, sodium bicarbonate (CRPMI), and 12% autologous serum to facilitate adherence. The cells were placed in an ultra-low adherence plate (Corning, Kennebunk, ME, USA) for 4 h at 37 °C and 5% CO_2_. Non-adherent cells were removed by three washes with prewarmed PBS, and adherent monocytes were cultured in CRPMI with 12% autologous serum for 12 days until their differentiation into macrophages. To confirm that the cultured cells were indeed macrophages, we verified the presence of the CD14 receptor using a CD14-FITC antibody (Mylteny Biotec, Bergisch Gladbach, DE, USA) and stimulated the cells with 300 ng/mL of LPS (Sigma Aldrich, Saint Louis, MO, USA), 30 µg/mL of Poly I:C (Santa Cruz, Dallas, PA, USA), and 30 µg/mL of zymosan (Sigma Aldrich, Saint Louis, MO, USA) to analyze NO production. Macrophage NO production was evaluated from the nitrite accumulation in the culture supernatant using the Griess reaction assay (Griess Reagent System, Promega, Madison, WI, USA) according to the manufacturer’s specifications. We generated a reference sodium nitrite curve based on serial dilutions and interpolated the absorbance of samples obtained from macrophages activated with the different ligands. A Multiskan^TM^ Go Microplate Spectrophotometer (Thermo Scientific^®^, Waltham, MA, USA) at 550 nm absorbance was used to measure reaction coloration.

### 2.4. Infection with M. bovis and Treatment with CFPE

Macrophage monolayers containing 3 × 10^5^ cells were infected with *M. bovis* (MOI, 10:1) and incubated for 4 h at 37 °C under 5% CO_2_ to allow phagocytosis. Then, the cells were washed four times with 5 mL of fresh CRPMI to remove extracellular bacteria and incubated with CRPMI with 10% heat-inactivated fetal calf serum (FCS) for different periods depending on the assay. We carried out a time-course study including 1, 2, 4, 8, 12, 16, and 24 h for the IL-1β secretion experiment. After two repetitions, we identified that, compared to 4 h, 8 h did not show any difference, and 12 h only showed a slight increase. Therefore, we decided to measure IL-1β and lactate dehydrogenase (LDH) release in the supernatant at 1, 2, 4, 16, and 24 h after incubation. Conversely, NLRP3 inflammasome inhibition, which decreased the rate of macrophage cell death, was measured after 4 h, whereas the IL-1β inhibition assay, the bactericidal assay used to estimate the intracellular growth of the mycobacterium, the western blot used for gasdermin D detection, and the detection of ASC specks were performed after 24 h. The supernatant was recovered after the infection time. All components derived from infection with *M. bovis* were sterilized by filtering and seeded to ensure they did not contain viable mycobacteria. Macrophage monolayers containing 1 × 10^6^ cells were incubated with 100 µg/mL of CFPE (1, 2, 4, 16, and 24 h for the LDH kinetic and propidium iodide (PI)/annexin V assays; 24 h for evaluation of IL-1β and detection of ASC specks). For the inflammasome inhibition assays, the chemical inhibitors (10 µM CRID 3, an NLRP3 inhibitor, and 50 µM Y-VAD, a caspase-1 inhibitor VI) (Sigma, Sant Louis, MO, USA) were added 2 h before *M. bovis* infection or CFPE treatment and kept in the culture medium throughout the incubation. In the cell death inhibition assays, the chemical inhibitors (10 µM CRID 3; 50 µM Y-VAD; 50 µM Necro-1, a RIP-1 inhibitor; and 50 µM Z-VAD, a pan-caspase inhibitor) (Sigma Aldrich, Saint Louis, MO, USA) were also added 2 h before *M. bovis* infection or CFPE treatment and kept in the culture medium throughout the incubation.

### 2.5. Quantification of Cell Death Types Using Annexin V/Propidium Iodide (PI)

We used a cell death apoptosis kit to quantify apoptotic and necrotic cells (Invitrogen, Eugene, OK, USA). Macrophages treated with CFPE were detached from the ultra-low adherence plate by pipetting, washed thrice with CRPMI, and kept suspended in binding buffer (10 mM HEPES/NaOH pH 7.4, 150 mM NaCl, 5 mM KCl, 1 mM MgCl2, 1.8 mM CaCl_2_) for 10 min. Subsequently, 1 µL of propidium iodide and 5 µL of Annexin V-FITC were added to 100 µL of binding buffer. An aliquot of 3 × 10^5^ cells was incubated with 100 µL of binding buffer at room temperature for 15 min. The cells were analyzed by cytometry, and the fluorescence emission was measured at 530 and 575 nm in an FACS Aria llu (BD, Franklin Lakes, NJ, USA) cytometer. The data were analyzed with FlowJo V10 software (FlowJo LLC, Ashland, OR, USA). We used 5 µM staurosporine (Sigma Aldrich, Saint Louis, MO, USA) for 24 h to induce apoptosis, 50 µM H_2_O_2_ for 4 h to induce necrosis, and 300 µg/mL LPS + 50 µM nigericin for 24 h to induce pyroptosis.

### 2.6. Detection of Lactate Dehydrogenase (LDH)

We evaluated necrosis by quantifying LDH release using the Cyto96^®^ Kit Cyto96^®^ (Promega Labs, Madison, WI, USA). The supernatant of the cells stimulated with CFPE or infected with *M. bovis* was collected at different times to measure LDH. For this, 50 µL of supernatant was added to 50 µL of LDH reagent (12 mg/mL lactate, 0.66 mg/mL iodine-nitro tetrazolium chloride, 4.5 units/mL diaphorase, 0.01% BSA, and 0.4% sucrose in PBS). The mixture was incubated at 37 °C for 20 min, and 50 µL of stop solution was added. The absorbance was read at 490 nm using the ELISA reader in a Multiskan^TM^ Go Microplate Spectrophotometer. Cells were lysed with a buffer (included in the kit) to determine the total LDH value (100% LDH), and the percentage of released LDH was estimated considering this value.

### 2.7. IL-1β Secretion

To evaluate IL-1β secretion, we analyzed the macrophage culture supernatant using an ELISA kit (Bovine IL-1β ELISA kit, Invitrogen, Vienna, Austria). Per the manufacturer’s instructions, we established a standard IL-1β curve by measuring the optical density with the Multiskan^TM^ Go Microplate Spectrophotometer reader at 550 nm and subtracting the values from the 450 nm wavelength measurements. We then subtracted the values corresponding to a 0 pg concentration of IL-1 (blanks) and determined the sample IL-1β concentration from the standard curve. LPS (300 ug/mL) was used as a positive control.

### 2.8. Detection of ASC Specks

Macrophages were seeded at a concentration of 3 × 10^5^ cells on chambered coverslips. The cells were either stimulated with CFPE 100 µg/mL or infected with *M. bovis* MOI 10:1 for 24 h; LPS 300 ng/mL was used as a positive control. In the inhibition treatments, inflammasome inhibitors were applied 2 h before receiving the corresponding stimulus. The cells were fixed with 4% paraformaldehyde and permeabilized with 0.15% Triton 100X for 10 min on ice. The cells were washed and incubated with the anti-TMS1/ASC polyclonal antibody ab15449 (Abcam, Cambridge, UK) at a dilution of 1:1000 at 4 °C overnight. Subsequently, they were incubated with the secondary antibody (Donkey anti-Goat IgG H&L-FITC, ab6881, Abcam, Cambridge, UK) diluted 1:3000 for 1 h in the darkness. The slides were washed and mounted using Vectashield^®^ medium with DAPI (Vector Laboratories, Burlingame, CA, USA). The images were acquired using an Olympus BX41 fluorescence microscope (Shinjuku, Japan) using 100× and 40× objectives. Images were captured with the Zen 2.6 blue edition software (Carl Ziess, Jena, Germany). The same capture parameters, exposure time, and intensity were applied for both the control and experimental samples in each staining system.

### 2.9. Analysis of Cleaved Gasdermin D in Granulomas from Cattle Naturally Infected with Mycobacterium bovis

We used formalin-fixed paraffin-embedded (FFPE) blocks of tissues (mediastinal lymph nodes) from six cattle previously characterized as naturally infected by *M. bovis*. For immunohistochemistry, 4–5 µm sections were obtained, placed on electrocharged glass slides, and deparaffinized. Heat-mediated antigen retrieval was performed with sodium citrate at 120 °C for 30 min (Diva Decloaker, 20X pH = 6). The slides were rinsed in TBS/Tween 0.01%, and endogenous peroxidase activity was blocked with 3% H_2_O_2_ for 10 min. The sections were washed in TBS/Tween 0.01% and blocked with Background Sniper (Biocare Medical, Concord, CA, USA) for 10 min. The samples were incubated overnight at 4 °C with a 1:100 dilution of cleaved gasdermin D monoclonal antibody (Asp275) (E7H9G) (Cell Signalling, Danvers, MA, USA). To detect the mycobacteria, we used the polyclonal antibody against *Mycobacterium tuberculosis* (TB) Concentrate # CP140 AC (Biocare Medical). The slides were rinsed again in TBS/Tween 0.01% and incubated with a Universal HRP-Polymer (Biocare Medical MRH538L10) for 30 min. Finally, the sections were incubated for 3 min with 3,3′-diaminobenzidine (DAB Substrate Kit, Peroxidase; Biocare Medical) and counterstained with Harris hematoxylin. Two to four sections of mediastinal lymph nodes per bovine were analyzed. A total of 56 granulomas were identified, and protein expression was quantified in the tissues. As a negative control, we used sections of the mediastinal lymph node of a bovine that showed no lesions suggestive of tuberculosis and were negative to *M. bovis* as tested by isolation and PCR. Digital images of the tissues were obtained at 40× magnification using a scanning microscope (Aperio Scanscope CS, Aperio, CA, USA). Staining was quantified by the Aperio Positive Pixel Count Algorithm count using the Aperio ImageScope system (Leica Biosystems, Wetzlar, DE, USA). The positive pixels were divided by the total area analyzed (mm^2^). Three different areas were selected: (a) areas with granulomas, (b) areas surrounding the lesion, and (c) tissue without infection, which was used as a negative control. The images presented are representative of our experimental findings.

### 2.10. CFU of M. bovis after Cell Death Induction

A total of 1 × 10^4^ cells per well were infected with *M. bovis* AN5 strains in Nunc MiniTrays (Nalge Nunc International, Rochester, NY, USA) for 4 h at an MOI of 10:1 per macrophage and maintained in a humidified atmosphere consisting of 5% CO_2_ at 37 °C. After allowing phagocytosis for 4 h, the cells were washed five times with 10 µL of CRPMI with 10% of FCS. The cells were treated with 5 µM staurosporine to induce apoptosis, LPS 300 µg/mL + nigericin 50 µM to induce pyroptosis, or CRPMI with 10% of FCS only (control). The initial count of the mycobacterial uptake was quantified by plating serial dilutions of the cell suspension after lysis with 0.5% Tween 20 (0 h). Intramacrophage mycobacterial growth was assessed 24 h after the infection, The supernatant of each well was recovered and the macrophages were lysed, the contents of the lysate were combined with the recovered supernatant for the complete evaluation of the mycobacteria (inside the macrophages and outside them). The count of the mycobacterial uptake was quantified by plating serial dilutions of the cell suspension. Colony-forming units (CFU) of *M. bovis* were determined by plating serial dilutions of the cell lysis suspension onto Middlebrook 7H11 medium with 10% OADC after 18 days of culture at 37 °C. The reported results are the average of three independent experiments.

### 2.11. Western Blot

Infected macrophages (3 × 10^5^) were lysed with Laemmli buffer; cell proteins were separated by SDS PAGE 10% and transferred to PVDF membranes. Gasdermin D was detected using a primary monoclonal antibody (Asp275) (E7H9G) (Cell Signaling, Danvers, MA, USA) and secondary antibodies anti-rabbit/HRP (Cell Signaling). The control load was monitored with anti-β actin (Cell Signaling, Danvers, MA, USA). The blots were incubated overnight with the primary antibodies diluted 1:1000, washed thrice, and incubated for 1 h with secondary antibodies diluted 1:10 000 and enhanced chemoluminescent (ECL) substrates (Femto West Super Signal Pierce, Rockford, IL, USA). Densitometry analysis was performed using ImageJ software 1.53t (National Institutes of Health, Bethesda, MD, USA). The results were calculated as the ratio protein of interest/control load and expressed as a fold increase in relation to the negative control. This experiment was repeated twice.

## 3. Results

### 3.1. M. bovis and Its CFPE Induce Necrotic Cell Death in Bovine Macrophages

The cells used in this study were confirmed to be macrophages since 91% of the cells expressed the CD14 receptor (Appendix A). The Griess assay showed that the stimulation with 300 ng/mL LPS, 30 µg/mL Poly I:C, and 30 µg/mL Zimosan significantly increased the production of nitrites from the baseline (5 µM) to 23 µM, 18 µM, and 15 µM, respectively (Appendix A).

The flow cytometry assay using Annexin V/PI stimulated with the CFPE showed a time-dependent decrease in live cells (Annexin V−/PI−) beginning at 70% and decreasing to 40% at 18 h and a time-dependent increase in necrotic cells (Annexin V−/PI+ and Annexin V+/PI+) beginning at 22% and reaching 55% at 18 h (Figure 1A). The PI+ area (considered necrotic cells) increased from 22% to 40%, starting at 4 h (Figure 1B). Regarding the LDH release, the CFPE-stimulated cells had significant damage from necrosis beginning at 2 h (45%), and the cells in necrosis peaked (77%) at 24 h post-treatment (Figure 1C). Infection with *M. bovis* also induced necrotic cell death in the cultured macrophages starting at 4 h post-infection (45% of the LDH release) and reaching maximum necrosis (75% of the LDH release) after 24 h (Figure 1D).

### 3.2. M. bovis and CFPE Induce IL-1β Release in Bovine Macrophages

To describe the IL-1β release pattern in our model, we stimulated the macrophages with LPS and quantified the IL-1β concentration in the supernatant after 2, 4, 16, and 24 h. In the macrophages stimulated with LPS (control), IL-1β was produced after 16 h (650 pg/mL) and reached its maximum level after 24 h (3200 pg/mL; Figure 2A). The cells stimulated with the CFPE for 24 h produced 1850 pg/mL, 3000 pg/mL, and 3200 pg/mL of IL-1β when exposed to 25 µg/mL, 50 µg/mL, and 100 µg/mL of CFPE, respectively (Figure 2B). The concentration of 100 µg/mL was selected, and the time course of IL-1β in the CFPE-stimulated cells (at 2, 4, 16, and 24 h) was similar to that produced by the LPS exposure; IL-1β production began after 16 h with a concentration of 500 pg/mL and increased to 2900 pg/mL at 24 h (Figure 2C). In the bacteria-infected macrophages, IL-1β was detected after 2 h (500 pg/mL) and reached its maximum concentration (3700 pg/mL) after 16 h (Figure 2D). Thus, both *M. bovis* and the CFPE induced the release of IL-1β.

### 3.3. IL-1β Release Depends on ASC and Caspase-1 in Macrophages Infected with M. bovis AN5

The presence of ASC specks suggests that IL-1β production was mediated by the inflammasome through the action of the ASC adapter protein and caspase-1. The proportion of macrophages containing ASC specks was 35% in the cells stimulated with the CFPE, 43% in the cells infected with *M. bovis*, and 5% in the unstimulated cells (Figure 3A). To determine the next step in the canonical pathway, we inhibited caspase-1 with the specific inhibitor Y-VAD, and IL-1β production decreased from an initial value of 3200 pg/mL to <300 pg/mL in macrophages treated with 250 µM and 50 µM Y-VAD and to 800 pg/mL in macrophages treated with 5 µM Y-VAD (Figure 3B). We thus used a Y-VAD concentration of 50 µM to perform the following inhibitions. After the Y-VAD-treated cells were incubated with the CFPE or *M. bovis*, the IL-1β concentration decreased from 3700 pg/mL to 470 pg/mL in the CFPE-incubated cells and from 3000 pg/mL to 2300 pg/mL in the *M. bovis*-incubated cells (Figure 3C). Thus, the inflammasome assembly and caspase-1 are involved in the release of IL-1β in the bovine macrophages stimulated with the CFPE and with *M. bovis*. One-way ANOVA showed a significant difference between the control versus the treated cells (*p* < 0.0001).

### 3.4. The NLRP3 Inflammasome Was Activated by M. bovis

We incubated bovine macrophages with diverse concentrations of CRID3 (1, 10, and 100 µM) and stimulated them with LPS 2 h later. The highest inhibitory effect was achieved using 10 µM and 100 µM CRID3, which decreased the production of IL-1β from 3200 pg/mL to less than 300 pg/mL (Figure 4A). Hence, we decided to use the 10 µM dose for the following experiments aimed at inhibiting the assembly of the NLRP3 inflammasome. Cells treated with CRID3 decreased their IL-1β concentration from 3200 pg/mL to 750 pg/mL in the CFPE-stimulated cells and from 3000 pg/mL to 1220 pg/mL in the *M. bovis*-infected macrophages (Figure 4B). The formation of ASC specks also decreased after the 10 µM CRID3 treatment, from 35% to 8% in the CFPE-stimulated cells and from 42% to 21% in the *M. bovis*-infected cells (Appendix A).

The relationship between necrosis and NLRP3 inflammasome activation was tested by inhibiting different inflammasome components. The inhibition of NLRP3 oligomerization with CRID3 and caspase-1 activation with Y-VAD along with CFPE or *M. bovis* stimulation resulted in less LDH release, decreasing significantly from 56% to 43% and from 50% to 27%, respectively (Figure 5A,B). Other types of cell death were tested with the chemical inhibitors Z-VAD, a pan-caspase inhibitor, and necrostatin-1 (Necro-1), an RIP1 inhibitor, but these did not diminish the LDH release. None of the chemical inhibitors affected the viability of uninfected macrophages (Figure 5C). Our results suggest that the inflammasome NLRP3 partially induces necrotic cell death. To support these data, we evaluated the presence of cleaved gasdermin D in the *M. bovis*-infected macrophages and CFPE-stimulated macrophages with LPS + nigericin. We observed a positive signal for gasdermin D in the three treatments 24 h after the stimulus, as measured by western blot (Appendix A).

### 3.5. Cleaved Gasdermin D Was Present in Granulomatous Lesions of Cattle Naturally Infected with M. bovis

We performed immunohistochemistry for gasdermin D using the cleaved gasdermin D (Asp275) antibody in different tissues with granulomatous lesions from cattle naturally infected with *M. bovis.* The lesions were positive for this protein at the edge of the granuloma necrosis and in the cells adjacent to the granulomas. The staining was diffuse, mainly in the cytoplasm of the macrophages and giant multinucleated cells. More bacilli were also observed in these cells (Appendix A). Moreover, we observed positivity to gasdermin D and mycobacterium debris outside the granulomas (Appendix A). The digital image analysis showed that gasdermin D was present in sections of the mediastinal lymph nodes from 6 cattle, in which 56 granulomas were analyzed. These tissues showed a higher gasdermin D positivity in the granulomas compared to the surrounding tissue and the uninfected tissue (negative control) (Figure 6).

### 3.6. Inflammasome Activation in Macrophages Decreases Intracellular CFU of M. bovis

Apoptosis induction with staurosporine 5µM decreased the mycobacterial growth in the macrophages from 5.2 × 10^5^ CFU (SD ± 1.6 × 10^5^) to 2.7 × 10^5^ CFU (SD ± 1.7 × 10^5^). Necrotic cell death induction with LPS (300 µg/mL) and nigericin (50 µM) decreased the mycobacterial growth from 5.2 × 10^5^ (SD 1.6 × 10^5^) to 0.8 × 10^5^ (SD ± 0.3 × 10^5^) (Figure 7A). Thus, both cell death inducers decreased the viability of the mycobacteria (Figure 7B), demonstrating that bacterial growth was inhibited by cell death and not by the chemical inducers per se. The induction of cell death with LPS and nigericin was evaluated using the LDH assay; stimulating bovine macrophages with LPS + 10 µM and 50 µM nigericin increased the LDH release from 23% to 58% (10 µM nigericin) and 81% (50 µM nigericin) (Appendix A).

## 4. Discussion

In this study, we evaluated the participation of the NLRP3 inflammasome and necrotic cell death in bovine macrophages infected with *M. bovis* or stimulated with components of the CFPE. Our main findings indicate that both stimuli induced the inflammasome assembly and pyroptosis-like cell death. The NLRP3 inflammasome is assembled through the canonical activation pathway with the participation of ASC and caspase-1 and favors the production of IL-1β. Necrosis in macrophages started as early as 4 h after bacteria or CFPE stimulation. It is worth highlighting that cell death decreases the viability of the mycobacteria and that the granulomas of cattle naturally infected by *M. bovis* showed cleaved gasdermin D. Together, these findings suggest that the pathology of bovine tuberculosis involves inflammation and necrosis caused by the inflammasome (Figure 8).

Mycobacteria can modulate the immune response and manipulate eukaryotic cell death. In antigen-presenting cells, apoptosis is associated with mycobacterial infection control, while necrosis is associated with their dissemination and survival, the type of cell death that occurs is related to the strains of mycobacteria and the characteristics of the infected cell; however, it is possible to find the presence of both types of cell death (apoptosis and necrosis) [14,15,16]. Previous studies from our group demonstrated that *M. bovis* induces caspase-independent apoptosis in bovine macrophages [12,17,18], but there were no studies describing the mechanisms of necrosis induction in the bovine model. In this study, we found that *M. bovis* can cause necrosis as a mechanism of cell death, which could explain the tissue damage observed in the infected bovines.

Necrotic cell death favors mycobacterial survival and dissemination and can be induced by viable bacteria or secreted proteins [19,20]. We found that both CFPE and viable mycobacteria induce necrotic cell death by rapidly damaging the cell membrane and allowing permeabilization to PI and LDH efflux, which are hallmarks of necrosis. The induction of this type of cell death was fast (within 4 h) in cells stimulated with the CFPE or infected with *M. bovis. M. tuberculosis* and *M. bovis* have virulence mechanisms that do not require viable bacteria to induce necrosis but are rather driven by secretion proteins. These proteins, such as PPEG68, PE25/PPE41, PPE31, ESAT-6, CFE-10, and CpnT, generate damage and permeability in mitochondria, phagosomes, and lysosomes, causing cell death [19,20,21,22,23]. We identified the presence of Mpb70, Mpb83, and sodB/sodA (manuscript accepted for publication in Veterinaria Mexico OA, 2023), the CFPE used in this study, as proteins related to cell death induction processes. However, other CFPE proteins or macromolecules could also induce necrosis similar to mycobacteria depending on their concentrations.

Mycobacteria can induce diverse types of necrosis-like cell death, such as necroptosis [24], ferroptosis [25], and pyroptosis [16]. In THP1 cells, *M. bovis* induces pyroptosis characterized by the participation of the NLRP7 inflammasome, the release of IL-1β, and the cell death [26]. Although our data clearly show that both *M. bovis* and the CFPE induce IL-1β production in bovine macrophages, this occurred considerably faster when the cells were exposed to viable bacteria than when exposed to the CFPE. The most common route of IL-1β production is mediated by the inflammasome, which requires at least two stimuli to be activated. Viable mycobacteria can activate the inflammasome through toll-like receptors, the production of reactive oxygen species, and the recognition of bacterial genetic material [9,27,28]. Therefore, viable mycobacteria produce more stimuli that activate the inflammasome, both at the membrane and intracellular level; this would explain why the whole bacteria stimulate IL-1β secretion faster than the CFPE alone.

The activation of the inflammasome is mediated by the activation and oligomerization of key molecules, including apoptosis-associated speck-like protein containing a CARD (ASC) and caspase-1 [29,30]. ASC speck evaluation provides evidence of the inflammasome assembly [16]. ASC specks were observed and quantified in the macrophages 24 h after *M. bovis* infection or CFPE stimulation. The percentage of specks was significantly higher in both cases compared to the control; however, the *M. bovis*-infected macrophages had more specks than the CFPE-stimulated macrophages. This difference may be due to the mycobacteria’s capacity to activate the inflammasome more efficiently than the CFPE, as discussed previously for IL-1β production. Caspase-1 activity was evaluated using the specific inhibitor Y-VAD, demonstrating the participation of caspase-1 in the maturation and release of IL-1β in macrophages infected with *M. bovis*. The participation of caspase-1 and ASC together are indicators of the canonical activation pathway of the inflammasome. Thus, our data suggest that most IL-1β production occurs through the activation of the canonical pathway in bovine macrophages infected with *M. bovis,* although smaller amounts of IL-1β could be produced via pathways that involve caspase-independent interleukin maturation mechanisms [31].

Several inflammasomes can detect mycobacteria and induce the production of IL-1β. Some reports show that non-tuberculosis mycobacteria activate the NLRP3 inflammasome [32,33,34], whereas bacteria from the tuberculosis complex activate the NLPR3, NLRP7, or AIM2 inflammasomes [9,26,27,35]. In this study, we evaluated the participation of the NLRP3 inflammasome because it has previously been shown to be activated by different mycobacteria in different macrophage models. Inhibiting the NLRP3 inflammasome with the specific inhibitor CRID3 produced a significant decrease in IL-1β production and ASC speck formation upon exposure to the CFPE or viable *M. bovis*, which highlights the importance of this receptor for recognizing *M. bovis* and its secretion components. Taken together, these results strongly suggest that the NLRP3 inflammasome participates in the maturation of IL-1β in *M. bovis* infections.

Some studies directly associate inflammasome activation with cell death processes [36,37]. To test the relationship between the inflammasome components and the induction of cell death, we inhibited several types of regulated cell death using chemical compounds. We used Z-VAD as a general caspase inhibitor to inhibit apoptosis, necrostatin-1 to inhibit RIP1, CRID3, and Y-VAD to inhibit the inflammasome capacity, and we found that the inhibition of caspases and RIP1 did not affect the cell mortality in infected cells. Inhibiting the oligomerization of NLRP3 and caspase-1 significantly decreased necrosis among the infected or CFPE-stimulated macrophages, suggesting that caspase-independent mechanisms may also contribute to necrotic cell death. Indeed, results from different authors have confronted the concept that gasdermin cleavage is only carried out by inflammatory caspases, suggesting that other unknown proteases may also play a role. Other mechanisms may also induce cell death; for instance, there is evidence showing that apoptosis can be related to inflammasome activation via caspase-8 or mitochondrial stress and that necroptosis induction can activate the inflammasome by a change in the potassium concentration, by caspase-6, or directly by RIPK1/3, MLKL, or ZIPB2 [12,18,37,38]. The activation of the inflammasome can lead to the cleavage of molecules like gasdermin D, increasing the membrane permeabilization and leading to cell death [39].

Histological sections of granulomas from animals naturally infected with *M. bovis* displayed the markers of cleaved gasdermin D in macrophages and multinucleated giant cells from granulomas in the lymph nodes and in the tissue adjacent to the granuloma. Cleaved gasdermin D is essential to induce necrosis; therefore, the intracellular presence of this molecule in the granuloma may indicate the capacity of the mycobacteria to induce inflammasome-mediated necrosis, mirroring our findings in vitro. In a parallel study, we observed acid-fast bacilli in lymph node granulomas from cattle naturally infected with *M. bovis*. To evaluate the presence of mycobacteria, we performed immunohistochemistry (IHC) using a polyclonal anti-mycobacterium antibody. IHC detected not only the presence of bacillus but also cellular remains as vacuoles and cytoplasmic dust, possibly associated with cell debris generated by the processing and phagocytosis of mycobacteria. Interestingly, the cytoplasm of different types of cells outside the granulomas also had cytoplasmic dust immunolabeling [40], (Appendix A). This can be explained by secretion components of mycobacterium, such as resuscitation-promoting factors (Rpf), which play a crucial role in the reactivation and resuscitation of latent or non-replicating bacterial cells that would allow bacteria to be kept outside the granuloma [41,42]. The presence of cleaved gasdermin D in vivo is an interesting result that suggests the participation of the inflammasome in necrosis induction in *M. bovis* infection. However, more experiments should be performed under different conditions to confirm this finding. The type of cell death is thought to contribute to the control or dissemination of mycobacteria [14,19]. However, recent studies have described several types of regulated cell death with necrosis-like characteristics that have not been described in terms of intracellular mycobacterium growth. Therefore, we evaluated the intracellular survival of *M. bovis* in cells subjected to inflammasome-mediated necrotic cell death and compared them to cells under apoptosis or not undergoing cell death. The infected cells under staurosporine-induced apoptosis displayed significantly limited mycobacteria growth, which is consistent with the findings of several authors [14,43]. Moreover, the *M. bovis*-infected cells treated with LPS + nigericin also controlled intracellular growth efficiently. This result could be related to inflammasome activation, which induces necrotic cell death and causes damage to the cells but can contribute to controlling the mycobacterial infection. Inflammasome activity controls the growth of some intracellular bacteria via the formation of membrane pores with the protein gasdermin D, and cells with deficiencies in some inflammasome components allow mycobacterial growth [44,45,46]. Furthermore, the inflammatory cytokines induced by inflammasome activity (IL-1β and IL-18) are critical for controlling mycobacterial infections. However, inflammasome-inducing ligands in macrophages also promote the production of diverse microbicidal molecules that play a fundamental role in controlling intracellular bacterial growth, such as reactive oxygen species and nitric oxide [11,13,44,45].

In conclusion, this work shows that viable *M. bovis* and CFPE activate the NLRP3 inflammasome via the canonical activation pathway in bovine macrophages as the most important mechanism of IL-1β production and release. Inflammasome activity begins with NLRP3 receptor oligomerization and the recruitment of ASC and caspase-1, which, at the same time, promote the maturation of IL-1β. Inflammasome activation partially leads to the cleavage of gasdermin D, which promotes the release of IL-1β and contributes to necrosis, suggesting a pyroptosis process. This type of cell death generates cell damage, visible as necrosis, that leads to tissue damage in individuals. However, it could also be part of a drastic mechanism to control mycobacterial growth and dissemination. The results obtained in this study and previous research by our group suggest that inflammasome-induced necrosis is not the only cell death mechanism generated by the immune system during mycobacterial infection in bovines. Other mechanisms, such as caspase-independent apoptosis and other necrotic mechanisms, may also participate in mycobacterial infection and, at the same time, generate damage to the host. Similar phenomena have been observed in infections with other pathogens, such as SARS-CoV-2, *Salmonella typhimurium*, and *Listeria monocytogenes*, in which exacerbated inflammation causes PANoptosis, a phenomenon involving several simultaneous types of cell death, such as pyroptosis, apoptosis, and necroptosis [46,47].

## Figures and Tables

**Figure 1 cells-12-02079-f001:**
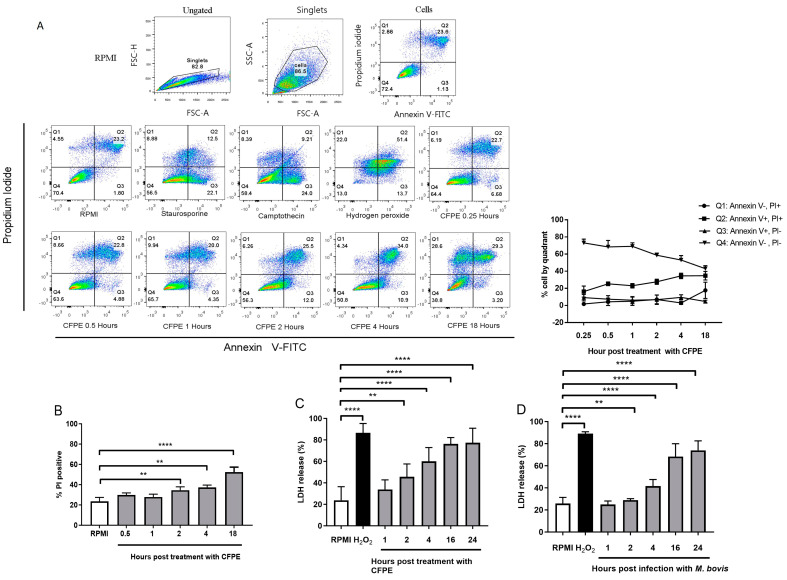
*Mycobacterium bovis* and its culture filtrate protein extract (CFPE) induce necrotic cell death in bovine macrophages. (**A**) Representative image of three experiments in which bovine macrophages were treated with 100 µg/mL of CFPE for different times (0.25, 0.5, 1, 2, 4, and 18 h), stained with Annexin V and propidium iodide. The control groups corresponded to 5 µg/mL staurosporine, 15 µg/mL camptothecin, and 50 µM H_2_O_2_. The upper part shows the gate history. (**B**) Percentage of necrotic cells (PI+) in macrophages treated with CFPE for different times (0.25, 0.5, 1, 2, 4, and 18 h). (**C**) LDH released in the supernatants of macrophages stimulated with CFPE at different times (1, 2, 4, 16, and 24 h). (**D**) LDH released in the supernatant of macrophages stimulated with *M. bovis* at different times (1, 2, 4, 16, and 24 h). Results are shown as the mean ± S.D. of three independent experiments with three internal replicas each. One-way ANOVA showed significant differences between the negative control versus treated cells. ** *p* value ≤ 0.01, **** *p* value ≤ 0.0001.

**Figure 2 cells-12-02079-f002:**
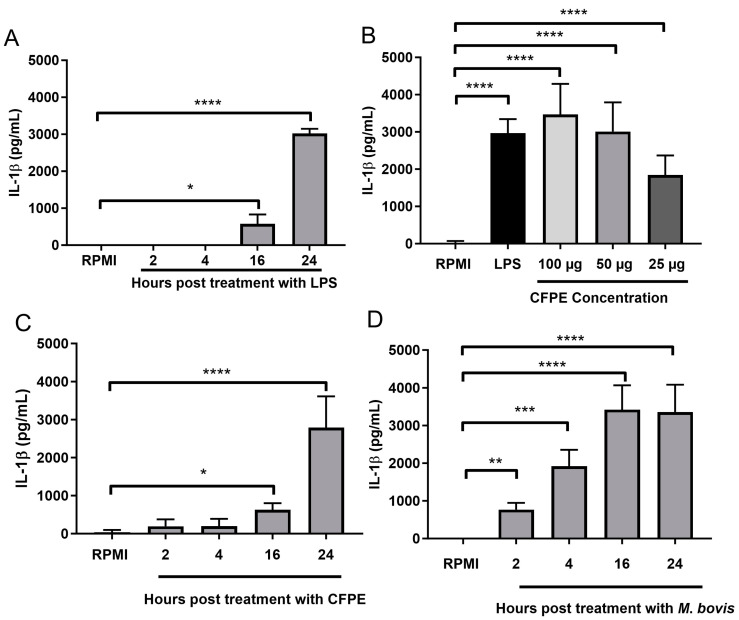
*Mycobacterium bovis* and its culture filter protein extract (CFPE) promote IL-1β release in bovine macrophages. (**A**) IL-1β production in bovine macrophages (3 × 10^5^) stimulated with LPS (300 ng/mL) at different times. (**B**) IL-1β production in macrophages (3 × 10^5^) stimulated with different concentrations of CFPE (25, 50, and 100 µg/mL). (**C**) IL-1β production in bovine macrophages (3 × 10^5^) stimulated with 100 µg/mL of CFPE/1 × 10^6^ cells at different times. (**D**) IL-1β production in bovine macrophages (3 × 10^5^) infected with *M. bovis* AN5 MOI 10:1 at different times. Results are shown as the mean ± S.D. of three independent experiments, each with three internal replicas. One-way ANOVA showed significant differences between the negative control and cells treated at different times or concentrations. * *p* value ≤ 0.05, ** *p* value ≤ 0.01, *** *p* value ≤ 0.001, **** *p* value ≤ 0.0001.

**Figure 3 cells-12-02079-f003:**
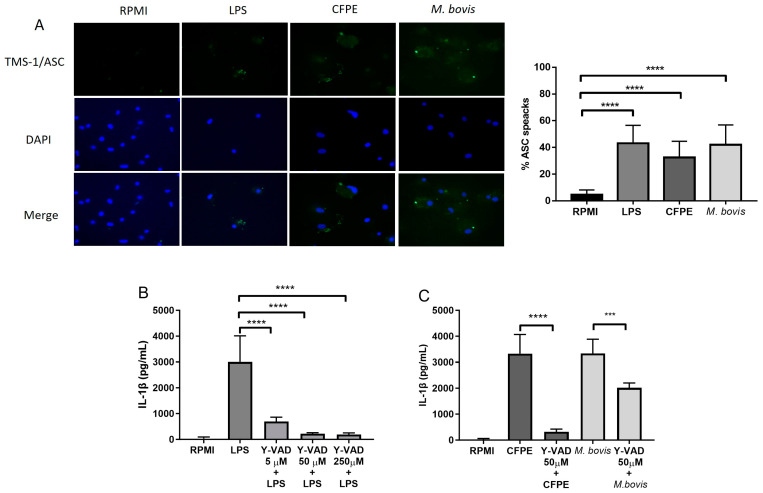
IL-1β release depends on ASC and caspase-1 in macrophages. (**A**) ASC specks in macrophages stimulated with LPS (300 ng/mL), culture filtrate extract (CFPE) at 100 µg/mL/1 × 10^6^ cells, or *M. bovis* MOI 10:1 for 24 h. Fluorescence microscopy with a 40× objective was used, and ASC specks were quantified in macrophages stimulated with LPS, CFPE 100 µg/mL in 1 × 10^6^ cells, or *M. bovis* MOI 10:1 for 24 h. ASC specks are shown in green, and the cell nucleus is shown in blue. (**B**) IL-1β production in bovine macrophages treated with 250, 50, and 5 µM of Y-VAD 2 h before LPS stimulation. (**C**) IL-1β production in bovine macrophages treated with Y-VAD (50 µM) for 2 h before stimulation with CFPE (100 µg/mL in 1 × 10^6^ cells) or *M. bovis* MOI 10:1, respectively. ASC speck percentage was calculated by counting more than 100 fields for each condition using fluorescence microscopy images with 40× magnification. Results are shown as the mean ± S.D. of three independent experiments, each with three internal replicas. One-way ANOVA showed significant differences between cells treated with LPS, *M. bovis,* or CFPE versus cells treated with Y-VAD more LPS, *M. bovis,* or CFPE, respectively. *** *p* value ≤ 0.001 and **** *p* value ≤ 0.0001.

**Figure 4 cells-12-02079-f004:**
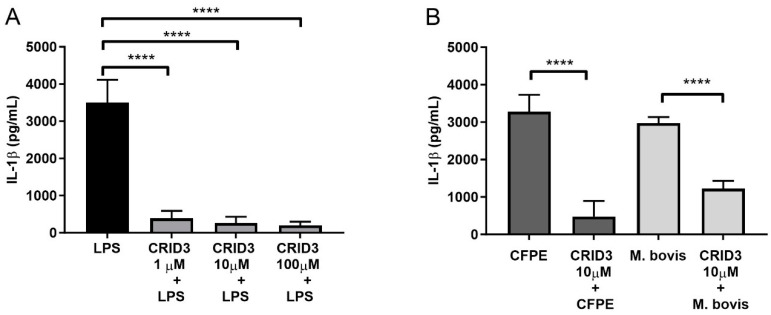
*M. bovis* activates the NLRP3 inflammasome. (**A**) IL-1β production in bovine macrophages treated with CRID3 (1, 10, and 100 µM) for 2 h before LPS stimulation (300 ng/mL). (**B**) IL-1β production in bovine macrophages treated with 10 µM CRID3 for 2 h before stimulation with 100 µg/mL of culture filtrate extract (CFPE) per 1 × 10^6^ cells or *Mycobacterium bovis* MOI 10:1. Results are shown as the mean ± S.D. of three independent experiments, each with three internal replicas. One-way ANOVA showed significant differences between cells treated with LPS, *M. bovis*, or CFPE versus cells treated with CRID3 and LPS, *M. bovis*, or CFPE, respectively. **** *p* value ≤ 0.0001.

**Figure 5 cells-12-02079-f005:**
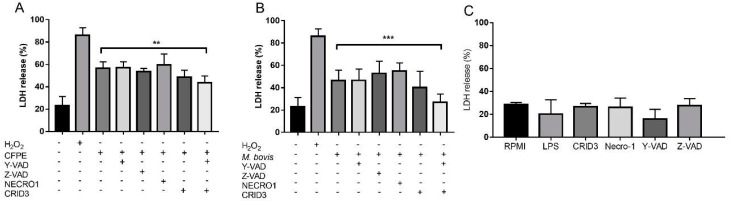
Inhibition of the NLRP3 inflammasome decreased the cell death rate of macrophages. (**A**,**B**) LDH in the supernatant of macrophages stimulated with 50 µm Y-VAD, 50 µm Z-VAD, 50 µm Necro-1, 10 µm CRID 3, and Y-VAD plus CRID3, 2 h before stimulation with 100 µg/mL of culture filtrate extract (CFPE) in 1 × 10^6^ cells or *M. bovis,* for 4 h. (**C**) LDH in the supernatant of macrophages stimulated with 50 µm Y-VAD, 50 µm Z-VAD, 50 µm Necro-1, 10 µm CRID-3, and Y-VAD + CRID-3. Results are shown as the mean ± S.D. of three independent experiments. One-way ANOVA showed a significant difference between the percentage of cell death in macrophages infected with *M. bovis* or CFPE versus macrophages infected with *M. bovis* or CFPE with cell death inhibitors (Y-VAD, Z-VAD, Necro-1, and CRID3). ** *p* value ≤ 0.01, *** *p* value ≤ 0.001. H_2_O_2_ or hydrogen peroxide a positive control, CFPE, Y-VAD, or Acetyl-tyrosine-valine-alanine-aspartate-chloromethyl ketone an irreversible inhibitor of caspase-1, Z-VAD or acetyl-tyrosine-valine-alanine-aspartate-chloromethyl ketone a pan-caspase inhibitor, Necro-1 or necrostatin-1 inhibitor of RIPK1 and CRID 3 an inhibitor of NLRP3 inflammasome.

**Figure 6 cells-12-02079-f006:**
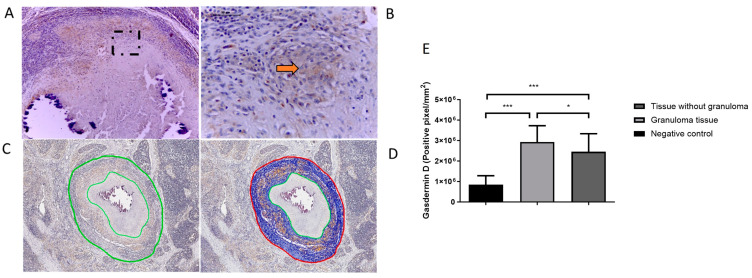
Quantification of cleaved gasdermin D staining in *Mycobacterium bovis*-induced granulomas in bovine lymph nodes. (**A**) Stage IV granuloma with a positive signal for gasdermin D at the periphery of the necrosis.(**B**) Amplification of the selected panel A (black box), the marks show positive cells with cleaved gasdermin D (indicated by arrow). (**C**) Selection of the lesioned area quantified by image analysis. (**D**) Number of positive pixels for each immunostaining was quantified using the ImageScope software system (version 12.1, Aperio, CA, USA). (**E**) Quantitative analysis of immunohistochemical staining. Results are shown as the mean ± S.D * *p* value ≤ 0.05, *** *p* value ≤ 0.001.

**Figure 7 cells-12-02079-f007:**
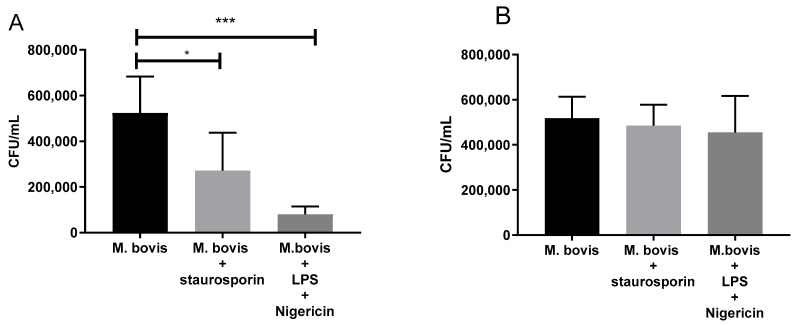
Activation of the inflammasome in macrophages decreases the intracellular survival of *Mycobacterium bovis*. (**A**) Bacterial CFU/mL of macrophages infected with *M. bovis* for 4 h, washed to eliminate non-phagocytosed bacteria, and incubated with staurosporine (5 µM) to induce apoptosis or with LPS (300 ng/mL) and nigericin (50 µM) to induce pyroptosis. The infected macrophages were lysed 24 h later, and the lysed cells were diluted and seeded on agar to calculate the CFU/mL. (**B**) CFU/mL of *M. bovis* exposed to staurosporine (5 µM) or LPS (300 ng/mL) and nigericin (50 µM) for 24 h. The exposed mycobacteria were recovered and seeded on agar to calculate CFU/mL. Results are shown as the mean ± S.D. of three independent experiments. Differences among treatments were tested with a one-way ANOVA. * *p* value ≤ 0.05, *** *p* value ≤ 0.0001.

**Figure 8 cells-12-02079-f008:**
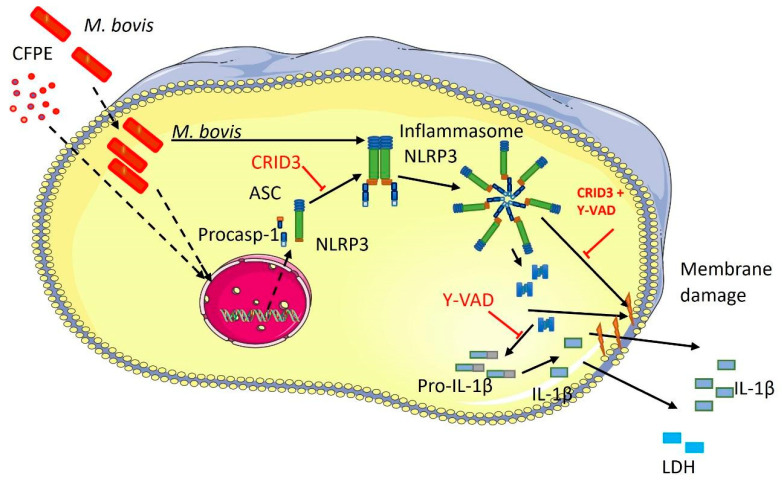
Proposed schematics of the NLRP3 inflammasome activation in infected macrophages. The diagram shows the ability of *Mycobacterium bovis* and its culture filtrate extract (CFPE) to activate the NLRP3 inflammasome, probably through pattern-recognition receptors. The assembly of the inflammasome leads to the maturation of IL-1β, which damages the cell membrane, resulting in necrotic cell death. This figure was created using the Servier Medical Art Commons Attribution 3.0 Unported Licence).

## Data Availability

The datasets generated in this study are available upon request from the corresponding authors.

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
