# Peer review of "Necrotic Cell Death and Inflammasome NLRP3 Activity in *Mycobacterium bovis*-Infected Bovine Macrophages"

_cells, 2023, doi:10.3390/cells12162079_

Round 1
Reviewer 1 Report
The peer-reviewed article investigates the effect of M. bovis on the necrosis activation on bovine macrophages in vitro. To fight infection, a clear understanding of the protection mechanisms of host is necessary. Macrophages are a key element of innate immunity in host defense against micobacteria. The relevance of investigation of M. bovis initiation/development in macrophages the different types of necrosis is no doubt. The methods used by the authors correspond to the tasks. However, there are no files with suppl. materials, which makes it impossible to review the article. As suppl. materials, a file with the main figures is attached.
Minor:
· line 126 - fetal calf serum (SFC) – standard abbreviation is FCS
· for figure 1A, a gating history should be given. This will make it easier to understand which cells are being examined.
· From the legend to Figure 2, as well as from the text in the results, it remains unclear whether the same or different number of cells were used in setting up different experiments. Please clarify it.
· Standard abbreviation for colony forming units is CFU. Please correct typos in text.
Author Response
Thank you for your observations, we hope to be able to review our manuscript, and we attach the response to the comments you made to us

Reviewer 2 Report
REVIEW:
GENERAL COMMENTS:
The article is well planned and written. Material and methods are carefully depicted.
ABSTRACT
Lines 20-21: “ASC” must be defined.
Lines 22-23: this paragraph has no verb.
INTRODUCTION
Lines 43-45: this affirmation might be accompanied by a reference.
MATERIAL AND METHODS
Specks = spots? Why apply this term in all the text?
Lines 133-134: the “bactericidal assay” either the western blot assay are not explained: Which is the purpose?
Lines 168-169: authors have developped other different ways to monitor the cell lysis degree.
Lines 232-234: Do authors think that 18 days is enough time to obtain colony growth after macrophage passage? Mycobacterial growth after intracellular passage is more difficult to obtain (lines 391-395).
RESULTS
Figure 5: is very complex to understand: H2O2, CFPE, Y-VAD, Z-VAD, NECRO1 and CRID can be explained in the text or in another table.
Figure 7: this experiment is very important to establish mycobacterial death and/or viability lost. Which is the difference between mycobacterial death and mycobacterial viability reduction according to these experiments? Colony count and culture incubation time to stablish this difference must be properly defined before. Is the same measurement system applied to distinguish between death and viability lost?
DISCUSSION
Figure 8: the diagram is very informative to understand these experiments.
Lines 429-435: Do both mechanisms (apoptosis and necrosis) can be activated at the same time? What is the final goal: mycobacterial reduction number and /or viability lost?
Lines 442-446: Do the authors identify some of the proteins outlined in line 444 in the culture filtrate?
Lines 513-522: Culture filtrate may contain many protein and non-protein factors to stimulate mycobacterial growth. In other words, mycobacterial growth and/or its viability may be independent of apoptosis or necrosis. Culture filtrate addition to a non-viable mycobacterial culture, can” revive” mycobacteria.
CONCLUSIONS
Conclusions are so adjusted to the experiments developped.
Author Response
Thank you for your comments, they enriched the manuscript considerably. We attach a file with the answer point by point

Reviewer 3 Report
The authors investigated the mechanism of bovine monocyte-derived macrophage (MDM) death following infection with Mycobacterium bovis or stimulation with secreted proteins from M. bovis cultures (CFPE). They find that bovine MDM rapidly activate the NLRP3 inflammasome to secrete IL-1beta and to trigger pyroptosis upon infection with live bacteria or stimulation with CFPE. Induction of pyroptosis with LPS+nigericin reduced the survival of intracellular M. bovis in macrophages.
Previously published papers provided evidence for NLPR3 inflammasome activation in mouse and human cells infected with M. bovis. However, this had not been examined bovine macrophages.
The mechanism and consequences of macrophage cell death due to mycobacterial infection are yet to be fully elucidated and may have a profound effect on the outcome of infection. The manuscript is interesting and well written, and the experiments are well designed. There are a few points that need to be clarified before the paper is ready for publication.
Comments:
1. The macrophages are dying very rapidly after M. bovis infection, which is different from what is observed in primary human macrophages infected with M. tuberculosis. Can the authors speculate why this might be. Is this a host or pathogen effect? A ratio of 10 bacteria /cell was used to infect the macrophages, based on CFU of the inoculum which gives an approximate value. Did the authors ever do an AFB stain to see what the average level of infection was on a cellular level? Could the level of infection be higher than thought which might explain the rapid onset of pyroptosis?
2. Alternatively, there may be differences in the host macrophages or mycobacteria species that accounts for this difference. A protein phosphatase PtpB secreted by M. tuberculosis inhibits macrophage inflammasome activation and pyroptosis in a gasdermin D-dependant manner and promotes intracellular survival of the bacteria (PMID: 36227980). Does M. bovis express this protein?
3. Section 2.10 – more detail is needed on the CFU method. Was the medium combined with the cell lysate for CFU determination or was it discarded. This is an important detail needed to understand the significance of the results in Fig 7. If the medium was included, it would indicate that the bacteria are being killed by the dying macrophages and that pyroptosis protects the host. If not then it may be the case that that bacteria are released into the medium, because pyroptosis is reducing their intracellular niche, and survive extracellularly which could favour dissemination.
4. Along the same lines, have the authors investigated the effect of inhibiting pyroptosis with CRID3 on M. bovis CFU?
Minor comments:
5. Line 82 What kind of Biosafety unit was used? Containment level 3?
6. Line 252 Correct the spelling of zymosan.
7. Lines 395 -7 The result is incorrectly described I think – presumably they mean to say that both cell death inducers did not directly affect the viability of M. bovis?
8. Line 452 – this refers to reference 37
9. Figure 1B: Correct Y-axis label to % PI positive
10. Figure 6E is incorrectly referred to as D in the legend.
Explain the difference between C and D with regard to the red and green highlighted areas.
11. Figure 7B - legend must be incorrect? i.e. presumably this is a macrophage-free (axenic) experiment?
- Y-axis of graphs – Label as CFU/ml
12. Fig S2 Graph X-axis, last bar - should it be labelled M. bovis + CRID3?
Author Response
Thank you very much for your comments and observations.We look forward to answering your concerns.

Round 2
Reviewer 1 Report
This manuscript could be recommended for publication as it is
Reviewer 3 Report
Minor point: The Y-axis label in Fig 1B still needs to be corrected to % PI positive.